# Chemical Composition and In Vitro Anti-*Helicobacter pylori* Activity of *Campomanesia lineatifolia* Ruiz & Pavón (Myrtaceae) Essential Oil

**DOI:** 10.3390/plants11151945

**Published:** 2022-07-27

**Authors:** Nívea Cristina Vieira Neves, Morgana Pinheiro de Mello, Sinéad Marian Smith, Fabio Boylan, Marcelo Vidigal Caliari, Rachel Oliveira Castilho

**Affiliations:** 1GnosiaH, Laboratório de Farmacognosia, Faculdade de Farmácia, Universidade Federal de Minas Gerais, Belo Horizonte 31270-901, Brazil; morganamello19@gmail.com; 2School of Pharmacy and Pharmaceutical Sciences, Trinity Biomedical Institute, Trinity College Dublin, Dublin 2, Ireland; fabio.boylan@tcd.ie; 3Departamento de Farmácia, Centro Universitário Santa Rita, Conselheiro Lafaiete 36408-899, Brazil; 4Department of Clinical Medicine, School of Medicine, Trinity College Dublin, Trinity Centre, Tallaght University Hospital, Dublin 24, Ireland; smithsi@tcd.ie; 5Departamento de Patologia Geral, Instituto de Ciências Biológicas, Universidade Federal de Minas Gerais, Belo Horizonte 31270-901, Brazil; caliari@icb.ufmg.br; 6Consórcio Acadêmico Brasileiro de Saúde Integrativa, CABSIN, São Paulo 05449-070, Brazil

**Keywords:** essential oil, chemical analysis, *Campomanesia*, *Campomanesia lineatifolia*, Myrtaceae, *Helicobacter pylori*, antibacterial activity

## Abstract

*Helicobacter pylori* is the most common cause of gastritis and peptic ulcers, and the number of resistant strains to multiple conventional antimicrobial agents has been increasing in different parts of the world. Several studies have shown that some essential oils (EO) have bioactive compounds, which can be attributed to antimicrobial activity. Therefore, EOs have been proposed as a natural alternative to antibiotics, or for use in combination with conventional treatment for *H. pylori* infection. *Campomanesia lineatifolia* is an edible species found in the Brazilian forests, and their leaves are traditionally used for the treatment of gastrointestinal disorders. Anti-inflammatory, gastroprotective, and antioxidant properties are attributed to *C. lineatifolia* leaf extracts; however, studies related to the chemical constituents of the essential oil and anti-*H. pylori* activity is not described. This work aims to identify the chemical composition of the EO from *C. lineatifolia* leaves and evaluate the anti-*H. pylori* activity. The EO was obtained by hydrodistillation from *C. lineatifolia* leaves and characterized by gas chromatography–mass spectrometry analyses. To assess the in vitro anti-*H. pylori* activity of the *C. lineatifolia* leaf’s EO (6 μL/mL–25 μL/mL), we performed broth microdilution assays by using type cultures (ATCC 49503, NCTC 11638, both clarithromycin-sensitive) and clinical isolate strains (SSR359, clarithromycin-sensitive, and SSR366, clarithromycin-resistant). A total of eight new compounds were identified from the EO (3-hexen-1-ol (46.15%), α-cadinol (20.35%), 1,1-diethoxyethane (13.08%), 2,3-dicyano-7,7-dimethyl-5,6-benzonorbornadiene (10.78%), aromadendrene 2 (3.0%), [3-S-(3α, 3aα, 6α, 8aα)]-4,5,6,7,8,8a-hexahydro-3,7,7-trimethyl-8-methylene-3H-3a,6-methanoazulene (2.99%), α-bisabolol (0.94%), and β-curcumene (0.8%)), corresponding to 98.09% of the total oil composition. The EO inhibited the growth of all *H. pylori* strains tested (MIC 6 μL/mL). To our knowledge, the current study investigates the relation between the chemical composition and the anti-*H. pylori* activity of the *C. lineatifolia* EO for the first time. Our findings show the potential use of the *C. lineatifolia* leaf EO against sensitive and resistant clarithromycin *H. pylori* strains and suggest that this antimicrobial activity could be related to its ethnopharmacological use.

## 1. Introduction

*Helicobacter pylori*, discovered by Marshall and Warren [1], is a Gram-negative spiral flagellated bacterium found in the stomachs of patients with chronic gastritis and peptic ulcers, and infects the stomachs of approximately half the global population [2]. The current treatment of *H. pylori* infection involves a combination of an antisecretory drug, such as proton pump inhibitors (PPI) or H_2_-blockers, together with two or three antimicrobials [3,4,5,6,7]. As expected for a multidrug therapeutic regimen, several side effects may be associated. In these cases, the patient may abandon treatment, contributing to the emergence of drug-resistant strains of *H. pylori* [8]. Bi et al. [9] demonstrated that herbal medicines are effective in treating gastric ulcers and have fewer side effects and lower recurrence rates. Moreover, the combination of herbal medicines and conventional anti-gastric ulcer drugs displays a synergistic effect against gastric ulcers [10,11]

In 2017, the World Health Organization listed clarithromycin-resistant *H. pylori* in the high priority category, in which prescribed treatment requires attention. The main point of this problem is that drug resistance has been markedly increasing and, consequently, the success of therapeutic regimens that include clarithromycin for eradicating *H. pylori* infection has been declining [12].

The genus *Campomanesia* is a member of the Myrtaceae family that comprises more than 100 genera and 4000 species around the world [13]. In Brazil, 41 species are known, of which 32 are endemic. These plants are represented by shrubs and trees found in the Brazilian Amazon rainforest, Atlantic forest, Cerrado and Caatinga, occurring in several regions of the country [14]. The *Campomanesia* species are popularly known as “guavira”, “guabirobeira”, or “gabiroba”, characterized by fruits with a citrus aroma and flavor that are used to prepare liqueurs, juices and sweets [15,16]. In addition, several species of *Campomanesia* are traditionally used for medicinal purposes, such as dysentery, stomach problems, diarrhea [17], cystitis and urethritis [18], as well as hepatic disorders [19]. Due to empirical knowledge of the health benefits associated with these species, the biological and pharmaceutical effects associated with *Campomanesia* have been the object of studies by different research groups [20,21,22,23].

When considering the antimicrobial activity, the essential oil obtained from *C. guazumifolia* showed strong inhibition for *Staphylococcus aureus*, *Escherichia coli* and *Candida albicans*, and demonstrated significant antioxidant activity [21]. In a study on the antimicrobial potential of essential oils from Cerrado plants against multidrug-resistant foodborne microorganisms, the essential oil from *C. sessiliflora* showed strong inhibition in clinical *Staphylococcus* strains, which cause bovine mastitis and are related to milk-borne diseases [22]. Furthermore, the antimicrobial and antibiofilm activities of the EO of the *C. aurea* showed sufficient capability against *Listeria monocytogenes* [23]. These results reveal the antimicrobial potential of the essential oils obtained from the *Campomanesia* species.

Although several studies showed that the plants and plant extracts/constituents exhibit anti-*H. pylori* activity and gastroprotective action, in recent years, few studies have described the effects of specific essential oils on *H. pylori* growth and viability [24,25,26].

Previous studies conducted by our research group demonstrated that *C. lineatifolia* Ruiz & Pavón leaf extracts protect the gastric mucosa against experimental ethanol- and indomethacin-induced gastric lesions [27]. Furthermore, the gastroprotective effects appear to be related to their antioxidant properties and polyphenolic contents. In a study of Brazilian medicinal plants in lipopolysaccharide (LPS)-stimulated THP-1 cells, Henriques et al. [28] demonstrated the in vitro tumor necrosis factor α (TNF- α) inhibitory activity of *C. lineatifolia*.

By considering a causal relationship between inflammation, gastric ulcers and *H. pylori* infection, we proposed a study to determine whether *C. lineatifolia* essential oil (EO) has anti-*H. pylori* activity. In the present study, we also aimed to identify the chemical composition of the EO that could be related to anti-*H. pylori* activity so that it can be used as an alternative or adjunctive agent to the current therapy for *H. pylori* infection. As far as we know, our study investigates, for the first time, the chemical composition and the anti-*H. pylori* activity from the essential oil of *C. lineatifolia*.

## 2. Results

### 2.1. Chemical Analysis of the Essential Oil

The GC-MS chemical analysis of *C. lineatifolia* essential oil showed the presence of eight components that represent 98.09% of the total oil composition (Table 1). The main components of the oil were 3-hexen-1-ol (46.15%), α-cadinol (20.35%), 1,1-diethoxyethane (13.08%), and 2,3-dicyano-7,7-dimethyl-5,6-benzonorbornadiene (10.78%), followed by aromadendrene 2 (3.0%), and [3-S-(3α, 3aα, 6α, 8aα)]-4,5,6,7,8,8a-hexahydro-3,7,7-trimethyl-8-methylene-3H-3a,6-methanoazulene (2.99%), respectively. The other components, such as α-bisabolol (0.94%) and β-curcumene (0.8%), were traced in the *C. lineatifolia* oil. The structures of the identified compounds are presented in Figure 1.

### 2.2. In Vitro Anti-Helicobacter pylori Activity Evaluation

The antimicrobial activity of the *C. lineatifolia* EO at 6 μL/mL–25 μL/mL concentrations was evaluated against ATCC 49503; NCTC116328 (both clarithromycin-sensitive strains), and SSR359; SSR366, clarithromycin-sensitive clinical isolate, clarithromycin-resistant clinical isolate, respectively. Clarithromycin was used as a positive control with inhibitory activity for the *H. pylori* growth in a concentration range of 0.03 μg/mL–256 μg/mL. It was shown that the EO was able to inhibit the growth of all the *H. pylori* strains at the lowest concentration tested, with a calculated MIC of 6 μL/mL. Clarithromycin inhibited the *H. pylori* growth at 0.03 μg/mL in the sensitive strains (ATCC 49503, NCTC11638, and SSR359) and at 8 μg/mL in the clarithromycin-resistant clinical isolate (SSR366).

## 3. Discussion

Essential oils are complex mixtures of volatile compounds belonging to various chemical classes, such as alcohol, ethers or oxides, aldehydes, ketones, esters, amines, amides, phenols, heterocycles, and mainly the terpenoids, especially monoterpenes and sesquiterpenes [36]. The chemical analysis of the EO composition from the *C. lineatifolia* leaves consisted mainly of alcohol (**2**) (46.15%), ether (**1**) (13.08%), and sesquiterpenes (**3**–**8**) (38.86%) as the majority constituents.

Osorio et al. [37] demonstrated that the presence of alcohols was predominantly characterized in the volatile extract of *C. lineatifolia* leaves, and among them, (Z)-3-hexenol was one of the majority constituents. In addition, cadinanes sesquiterpenes, β-cadinol, γ-cadinol, and δ-cadinol were identified. Similarly, in this study, the alcohol 3-hexen-1-ol (**2**) corresponded to the majority compound in the essential oil, and the cadinane sesquiterpene, α-cadinol (**8**), was also identified in *C. lineatifolia*.

Moreover, the compounds ether 1,1-diethoxyethane (**1**), 2,3-dicyano-7,7-dimethyl-5,6-benzonorbornadiene (**3**), [3*S*-(3α, 3aα, 6α, 8aα)]-4,5,6,7,8,8a-hexahydro-3,7,7-trimethyl-8-methylene-3*H*-3a,6-methanoazulene (**4**), aromadendrene 2 (**5**), β-curcumene (**6**), and α-bisabolol (**7**) were identified for the first time in *C. lineatifolia*.

The phytochemical analysis of the *Psidium guajava* leaves, Myrtaceae family, showed the presence of the ether 1,1-diethoxyethane (**1**) as one of the constituents that confer fruit aroma [38].

Compound **3**, 2,3-dicyano-7,7-dimethyl-5,6-benzonorbornadiene, was described in a study of the chemical composition of the essential oils isolated from the *Mentha microphylla* (Labiatae) and *Lantana camara* (Verbenaceae) species [39], with related insecticidal and fungicidal activities. This sesquiterpene was also identified in the dichloromethane-extracted rhizome essential oil from *Aframomum atewae* (Zingiberaceae), and the EO exhibited fungicidal activity against *Candida albicans* [40].

Compound **4**, [3*S*-(3α, 3aα, 6α, 8aα)]-4,5,6,7,8,8a-hexahydro-3,7,7-trimethyl-8-methylene-3*H*-3a,6-methanoazulene, showed an identical fragmentation profile to the diene derivative of the sesquiterpene zizanol, found in *Vetiveria zizanioides* (L.) Nash, sin. *Chrysopogon zizanioides* (L.) Roberty (Poaceae) [31]. Vetiver oil is commonly used in the perfume and food industry as a flavoring agent [41]. Aside from its special aroma, its antioxidant [42], antibacterial [43] and anti-inflammatory [44] activities are described in the literature.

Aromadendrene 2 (**5**) was illustrated in several species of the Myrtaceae family, such as *Eucalyptus* sp. [32,45], *Psidium guajava* [46], and *Lophomyrtus* sp. [47].

Compound **6**, β-curcumene, was identified as one of the major constituents in the essential oil from the rhizomes of *Curcuma amada* and *C. longa* (Zingiberaceae) [48,49]. *Curcuma* sp. is traditionally used to treat various intestinal diseases, fever and jaundice [50] and its anti-*H. pylori* activity has been described [51,52]. In the Myrtaceae family, γ-curcumene is found in the essential oil of *Eucalyptus microtheca* leaves [32] and β-curcumene in *Myrcia sylvatica* [53].

Compound **7**, α-bisabolol, was identified in the essential oils of some species from the Myrtaceae family, such as *Plinia cerrocampanensis* [54] and *Psidium myrtoides* [55].

The treatment for *H. pylori* infection commonly involves first-line triple therapy with clarithromycin (a combination of second-generation PPI, amoxicillin and clarithromycin for 14 days) [7]. In areas of high clarithromycin resistance (> 15%), a quadruple bismuth therapy (PPI, bismuth salt, tetracycline, nitroimidazole) or a concomitant therapy (PPI, clarithromycin, metronidazole, amoxicillin) is recommended [3,4,5]. However, these regimens may increase the risk of side effects and the cost of treatment, which brings difficulties for their implementation in clinical practice and often causes treatment abandonment by the patient. In addition, the use of multiple antimicrobial agents for *H. pylori* infection may increase the risk of future microbial resistance [56]. In this scenario, new regimens and approaches that allow the minimal use of antimicrobials for a shorter treatment period are needed.

Plant-based medicines are effective in treating gastric ulcers and have fewer side effects and lower recurrence rates, and when used in monotherapy or in combination with conventional drugs, they can become an alternative to treat certain gastric ulcers and prevent their recurrence [9]. Clinical studies have already demonstrated the efficacy of some natural products for the treatment of gastrointestinal diseases. For example, capsaicin (derived from the peppers of the *Capsicum* genus) protected the gastric mucosa in healthy volunteers who took acetylsalicylic acid [57] and decreased the intensity of dyspeptic symptoms in patients with functional dyspepsia [58]. Curcumin, from *Curcuma longa* (turmeric), improved dyspeptic symptoms, improved quality of life, and provided a satisfactory equivalent to omeprazole in a double-blind, placebo-controlled trial in patients diagnosed with functional dyspepsia [59]. Patients treated with a *Maytenus ilicifolia* (espinheira-santa) extract for 28 days showed a significant improvement compared to the placebo group, with regard to the overall dyspeptic symptoms, and especially for symptoms of heartburn and gastralgia [60]. 

Regarding the essential oils, Korona-Glowniak et al. [24] evaluated the in vitro activity of the five essential oils, silver fir, pine needle, tea tree, lemongrass and cedarwood oils, against *H. pylori*. The most active against the clinical strains of *H. pylori* were cedarwood oil and oregano oil. Moreover, the cedarwood oil inhibited the urease activity at sub-inhibitory concentrations, suggesting that this essential oil can be a useful component of the plant preparations supporting the eradication of *H. pylori* therapy. The *Pelargonium graveolens* oil exhibited good activity against *H. pylori* at a MIC of 15.63 µg/mL, and once combined with clarithromycin, a significant synergistic effect appeared at a fractional inhibitory concentration index of 0.38 µg/mL [25]. Thus, the combination of herbal medicines with anti-*H. pylori* activity could improve the outcome for gastric ulcer patients.

The alcohol, 3-hexen-1-ol, is described as one of the majority constituents of the EO of *Fagopyrum esculentum* (Polygonaceae) flowers and is related to the antimicrobial activity against Gram-positive bacteria *Bacillus subtilis* and *Staphylococcus aureus*, and Gram-negative bacteria, *Escherichia coli* and *Pseudomonas lachrymans* [61]. Pino et al. [62] also described the presence of 3-hexen-1-ol as the majority constituent of the EO from *Galinsoga parviflora* (Asteraceae) leaves and demonstrated that this EO exhibited antimicrobial activity against Gram-positive bacteria *S. aureus* and *Bacillus cereus*. Similarly, this alcohol corresponded with the majority constituent in the *C. lineatifolia* EO, and together with the data described in the literature, one can suggest its potential anti-*H. pylori* activity, observed in this study.

David et al. [63] demonstrated that the EO obtained by the hydrodistillation of *Vetiveria zizanioides* leaves showed potent antimicrobial activity against the Gram-positive bacterium *S. aureus* and moderate activity for the Gram-negative bacteria *P. aeruginosa* and *E. coli*. It is important to consider that *V. zizanioides* contains a diene derivative of the sesquiterpene zizanol, [3S-(3α, 3aα,6α,8aα)]-4,5,6,7,8,8a-hexahydro-3,7,7-trimethyl-8-methylene-3H-3a,6-methanoazulene [27], also identified in the current study for *C. lineatifolia* EO. Mulyaningsih et al. [64] identified that aromadendrene contributes to the antimicrobial activity of *Eucalyptus globulus* essential oil, markedly on multidrug-resistant bacteria, such as methicillin-resistant *S. aureus* (MRSA) and vancomycin-resistant *Enterococcus faecalis* (VRE), in addition to activity against the Gram-negative bacterium *Acinetobacter baumanii*. Furthermore, related to *Campomanesia adamantium* and *C. pubescens* species, in which aromadendrene and α-cadinol were identified, the hexanic extract of the fruits inhibited the growth of all bacteria tested, i.e., *S. aureus*, *P. aeruginosa*, and *E. coli*, demonstrating that the volatile compounds present showed good activity against Gram-positive and Gram-negative bacteria [65].

It has been shown that β-curcumene and α-bisabolol, the majority constituents of the EOs of *Curcuma* sp. (Zingiberaceae) and *Plinia cerrocampanensis* (Myrtaceae), respectively, have anti-*H. pylori* activities [46,47,49]. In addition, Brehm-Stecher & Johnson [66] described that bisabolol is able to increase bacterial wall permeability and, consequently, susceptibility to antibiotics of clinical importance, suggesting a possible mechanism for its activity.

Studies indicate that plant essential oils have bactericidal effects against *H. pylori*, and this activity could be related mainly to the terpenoid content, followed by other compounds, such as phenols, alcohols, aldehydes and ketones [24,26,36,67]. Additionally, the EOs have equivalent bactericidal effects against both the antibiotic-susceptible strains and other *H. pylori* strains that resist antibiotics, suggesting that EOs may be promising candidates to treat *H. pylori* infection [68].

All these findings related to the compounds identified in the EO of *C. lineatifolia* may suggest that the anti-*H. pylori* activity of the EO may originate from one of its main constituents or even synergism between them. However, it is important to note that the antibacterial assay was performed as an initial screen to determine whether the *C. lineatifolia* EO demonstrated anti-*H. pylori* activity. In the current study, the microbroth dilution assay data presented demonstrates the bacteriostatic effect of the EO at a MIC of 6 μL/mL, as *H. pylori* growth was inhibited at all concentrations of EOs tested (6 μL/mL–25 μL/mL). Furthermore, lower concentrations of EO (i.e., < 6 μL/mL) could possibly demonstrate anti-*H. pylori* activity, and will be the subject of our future MIC and minimum bactericidal concentration (MBC) studies after purifying additional EO from fresh *C. lineatifolia* leaves to determine whether the EO also possesses bactericidal activity.

These results showed the potential of the *C. lineatifolia* EO against clarithromycin-sensitive and resistant *H. pylori*-type cultures and clinical isolate strains, making the species promising in the treatment of gastric ulcers associated with *H. pylori* infection.

## 4. Material and Methods

### 4.1. Plant Material

*Campomanesia lineatifolia* leaves were collected in February 2020 from Minas Gerais, Brazil (19°52′9.87″ S; 43°58′12.04″ W). The species was identified by Dr Marcos Sobral from the Botany Department of Biological Sciences, the Institute at the Federal University of Minas Gerais (UFMG), Belo Horizonte. A voucher specimen (BHCB 150.606) was deposited at the UFMG Herbarium. The registration in the National System of Genetic Heritage and Associated Traditional Knowledge Management (SisGen) was carried out and given the code A216C7C.

### 4.2. Isolation of the Essential Oil

The EO was extracted from the fresh leaves of *C. lineatifolia* by hydrodistillation using a Clevenger-type apparatus. The extraction was carried out for 3 h by mixing 143 g of plants in 1000 mL of distilled water. The EO were dried with sodium sulphate anhydrous (Dinâmica Química Contemporânea Ltda), and concentrated under reduced pressure by a rotatory evaporator to evaporate water. The pure oil was stored at −25 °C until further analysed. The essential oils’ yield was 0.5 mL.

### 4.3. Analysis of the Essential Oil

The essential oil chemical composition assessments and the identification of the main constituents were conducted by GC-MS analyses. The GC-MS analysis was performed in triplicate using a Hewlett-Packard 6890N gas chromatograph, coupled with a 5975B mass selective detector (MSD; Agilent Technologies, Inc., Santa Clara, CA, USA) operating at 70 eV over a mass range of 35–500 amu and a scanning speed of 0.34 and equipped with a DB-5MS fused-silica capillary column (5% phenylmethylsiloxane; length—30 m, internal diameter—0.25 mm, film thickness—0.25 μm). The oven temperature was raised from 70 °C to 290 °C at a heating rate of 5 °C/min and held isothermal for 10 min; injector temperature, 250 °C; interface temperature, 300 °C; carrier gas, helium at 1.0 mL/min. The sample oil was dissolved in Et_2_O at the ratio of 1:1000 and injected into a pulsed split mode. The flow rate was 1.5 mL/min for the first thirty seconds (0.50 min), then modified to 1.0mL/min for the remainder of the run; the split ratio was 40:1. The identification of the constituents was based on the comparison of their linear retention indices relative to the series of n-hydrocarbons (C6–C40), and on computer matching against commercial (NIST or Wiley library) and home-made library mass spectra that were built up from pure substances and components of known oils, and MS literature data.

### 4.4. Anti-Helicobacter pylori Activity

#### 4.4.1. Bacterial Strains and Culture Conditions

The *H. pylori* strains were obtained from the American Type Culture Collection (ATCC 49503, clarithromycin-sensitive), National Collection of Type Cultures (NCTC11638, clarithromycin-sensitive), and from Tallaght University Hospital, Dublin 24, Ireland (SSR359, clarithromycin-sensitive clinical isolate, and SSR366, clarithromycin-resistant clinical isolate). All strains were maintained on Columbia blood agar (CBA) plates containing 5% sheep’s blood (VWR) at 37 °C under microaerobic conditions (CampyGen, Oxoid). Bacteria were inoculated into a brain heart infusion (BHI) broth (Sigma) containing 10% foetal bovine serum (Gibco) and incubated under microaerobic conditions at 37 °C with shaking (120 rpm) for 24h to prepare the bacterial suspensions for the broth microdilution assays.

#### 4.4.2. Broth Microdilution Assays

The determination of the minimum inhibitory concentration (MIC) of EO against *H. pylori* was evaluated using the broth microdilution assays in 96-well plates, according to the Standards for Antimicrobial Susceptibility Testing protocol [69], with adaptations [70]. The EO was diluted with the BHI broth supplemented with 10% foetal bovine serum and dimethylsulfoxide 2% (DMSO, Sigma) at 47 °C to prepare the concentrations at 6 μL/mL–25 μL/mL (0.6–2.5% *v*/*v*).

A total of 100 µL aliquots of overnight bacterial suspension (approximately 10^6^ CFU/mL; 1:20 0.5 McFarland Standard) [70] was added to each well of a 96-well plate. The cultures were incubated with 100 µL aliquots of EO for 72 h at 37 °C under microaerobic conditions (CampyGen) with shaking (120 rpm). The absorbances at a wavelength of 600 nm were read using a Varioskan LUX Microplate Reader (Thermo Fisher Scientific). The MIC values were determined as the lowest concentration at which no bacterial growth occurred. Clarithromycin was used as the positive control at 0.03 μg/mL–256 μg/mL (0.003–25.6% *v*/*v*).

### 4.5. Statistical Analysis

The experiments were performed in triplicate, and the results were expressed as mean ± SEM. A one-way ANOVA using GraphPad Prism 6.0, followed by a Bonferroni post-test, was performed, considering values of a *p* ≤ 0.05 as statistically significant. The MIC values were calculated considering the concentrations that did not differ statistically from the BHI broth control.

## 5. Conclusions

As far as we know, our study investigates, for the first time, the chemical composition of the essential oil from *C. lineatifolia* fresh leaves and its anti-*H. pylori* activity. Using the GC-MS technique, it was possible to identify eight new compounds for the species, representing 98.09% of the total oil composition, the major components of the alcohol 3-hexen-1-ol and the sesquiterpene α-cadinol.

Our findings demonstrate that the *C. lineatifolia* EO has anti-*H. pylori* activity at the lower concentration tested (MIC 6 μL/mL), suggesting a bacteriostatic property and a relation with its ethnopharmacological use as a medicinal plant for the treatment of gastric ulcers. Furthermore, the antibacterial activity against clarithromycin-resistant clinical isolates indicates that essential oils could be used in combination, as even antibiotics with high activity are not effective when used singly in therapeutic regimens involving patients with resistant *H. pylori* infection. Evidence for the use of medicinal plants in therapy shows that they have fewer adverse effects and lower rates of infection recurrence, as well as reducing time and improving the patient’s compliance with treatment.

For the follow-on work to this study, additional EO from fresh *C. lineatifolia* leaves will be purified to determine whether the EO also possesses bactericidal activity by performing MBC assays. In addition, it will be necessary to examine each antibacterial component of essential oil separately and in combination to ascertain whether they act alone or synergistically. Furthermore, studies involving in vivo models of gastric ulcers are proposed to determine the mechanisms by which the chemical constituents described here exert their anti-*H. pylori* activity.

## Figures and Tables

**Figure 1 plants-11-01945-f001:**
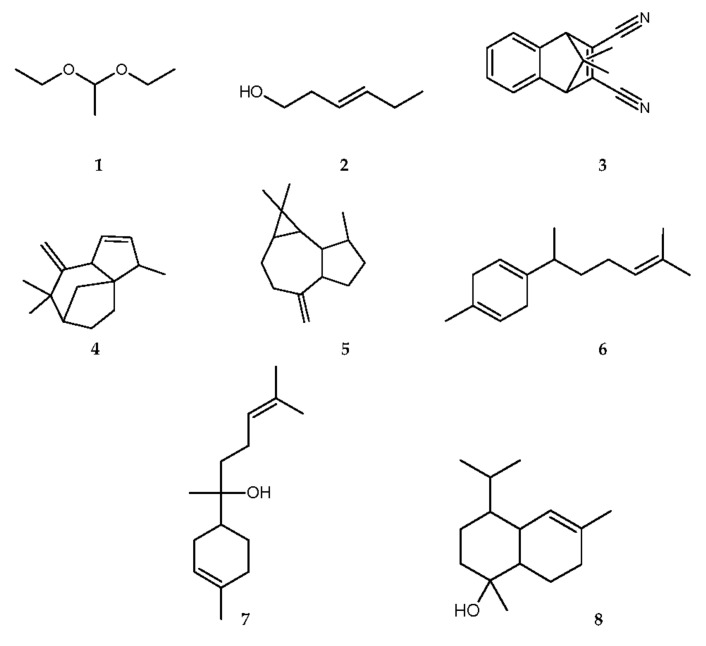
Structures of identified compounds (1–8) from *Campomanesia lineatifolia* fresh leaves essential oil.

**Table 1 plants-11-01945-t001:** Chemical compounds identified in essential oil from *Campomanesia lineatifolia* fresh leaves by gas chromatography–mass spectrometry. For methodology, see Material and Methods.

Compound	RT (min)	RI	Molecular Weight	Molecular Formula	%	MS	ID
1,1-diethoxyethane (**1**)	2.483	718	118	C_6_H_14_O_2_	13.08	103, 75, 73, 61, 47, 45, 43, 31, 29	NIST; Willner, Granvogl & Schieberle (2013) [29]
3-hexen-1-ol (**2**)	3.523	850	100	C_6_H_12_O	46.15	82, 79, 72, 69, 67, 65, 57, 55, 53, 51, 41, 39	NIST; Marongiu et al. (2001) [30]
2,3-dicyano-7,7-dimethyl-5,6-benzonorbornadiene (**3**)	18.158	1506	220	C_15_H_12_N_2_	10.78	223, 221, 207, 206, 205	Wiley
[3-*S*-(3α, 3aα, 6α, 8aα)]-4,5,6,7,8,8a-hexahydro-3,7,7-trimethyl-8-methylene-3*H*-3a,6-methanoazulene (**4**)	20.044	1586	202	C_15_H_22_	2.99	187, 159, 145, 131, 119, 105, 91, 77	Kang & Monti (1984) [31]
Aromadendrene 2 (**5**)	20.275	1596	204	C_15_H_24_	3.0	161, 133, 119, 105, 93, 91, 81, 79	NIST; Maghsoodlou et al. (2015) [32]
β-curcumene (**6**)	21.547	1652	204	C_15_H_24_	0.8	161, 119, 105, 93, 81, 55, 41	Wiley; Lucero, Estell & Fredrickson (2003) [33]
α-bisabolol (**7**)	21.555	1652	222	C_15_H_26_O	0.94	204, 161, 119, 105, 93, 69, 41	NIST; Zheng et al. (2004) [34]
α-cadinol (**8**)	21.809	1664	222	C_15_H_26_O	20.35	204, 189, 161, 134, 121, 109, 105, 95, 81, 71	NIST; Yu et al. (2004) [35]

RT: retention time; RI: retention index, determined relative to n-alkanes (C6–C40) on the DB-5MS column; %: relative percentage, obtained from peak area; MS: mass spectrum fragmentation; ID: proposed identification according to mass spectrum compared to NIST or Wiley library/literature and retention index, according to literature.

## Data Availability

All data generated or analyzed during this study are included in this article.

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
