# Peer review of "Chemical Composition and In Vitro Anti-Helicobacter pylori Activity of Campomanesia lineatifolia Ruiz & Pavón (Myrtaceae) Essential Oil"

_plants, 2022, doi:10.3390/plants11151945_

Round 1

Reviewer 1 Report

The authors describe a study about “Chemical Composition and In Vitro anti-Helicobacter pylori Activity of Campomanesia lineatifolia Ruiz & Pavón (Myrtaceae) Essential Oil. The topic is interesting but the experimental procedures have to be implemented.

MAJOR REVISIONS

Line 126: MIC has to be espressed as µg/mL and not as percentage. The minimum bactericidal concentration (MBC) assay has to be performed.

Line 308: specify the CFU/mL or CFU/well, the Authors used for the test. Furthermore the microdilution method is not well described.

MINOR REVISIONS

Line 23:  after the name “Campomanesia lineatifolia“  write the abbreviation (C. lineatifolia) and use it  in the text

Line 39 and 255: the Authors tested only sensitive and resistant strain to clarithromycin. So specify it.

Line 46 after the name  “Helicobacter pylori “ write the abbreviation (H. pylori). Delete “1984”. Write Gram with capital letter; it is the surname of Christian Gram

Line 53: “ delete (2014)

Line 91 : delete (2016)

Line 140; delete (2016)

Line 56: add a reference on synergistic action

Line 72 : the Authors wrote “…….pharmaceutical effects associated with Campomanesia have been the object of studies  by different research groups” but indicate only one reference. So, add more references or change the sentence.

Line 85: add references. Although few, other studies on EO have been published.

Lines 102,108,110, 123, 130 ecc: write the name of plants and bacteria in italic.

Reviewer 2 Report

This manuscript describes the chemical composition and in vitro anti-Helicobacter pylori activity of Campomanesia lineatifolia essential oil (EO). Despite, it seems to present some (few) data regarding this activity there are some issues that should be improved:

- lines 126, 130 and 314 the range should be from the minor to the higher value. 

- lines 131-132, the authors argue that the MIC values of the EO were achieved at the lowest concentration tested of 0.6%. Probably if the authors tested also lower concentrations the MIC was also achieved (from a point of view of methodological approach this seems that the concentration of the EO tested were not sufficient as it could be very important from which concentration it was achieved the MIC). The same happens for the clarithromycin with the sensitive strains. In addition, in the case of the EO the values were presented in % while in the case of clarithromycin, but it is not understandable if the values were higher or lower between them (probably it could be interesting to use the same units with some approximation for a better comparison). 

- lines 134-136 please remove this phrase as here it was the description of the results section.
- lines 209-214 should be inserted at the beginning of the first paragraph of discussion as the these description is partially repeated. In addition in the discussion between the third and fourth paragraph that describes some studies regarding this compound is missing the compound 2 activity and after the compound 7 is missing the description of compound 8 activity that were presented only at the end of the discussion. Maybe the interconnection between the topics in discussion is also needing to be more concise and organized (for ex. between lines 175-214 the information is more generalist regarding the information/literature that exists but this interrupted the sequence in which were described the activity of the different constituents).
- lines 252-254, this phrase is completely speculative, which analysis indicated that there is the synergism of the constituents of the essential oils? I think that this only can be assumed if the authors tested also the individual activity of each constituent of the EO.

- Along the manuscript, H. pylori and C. lineatifolia should be in italic and there are some language errors, for ex. lines 193, it should be curcumin (instead curcumina); lines 306 dimethylsulphoxide should be dimethylsulfoxide. Probably it could be needed a language revision by a native.

Round 2

Reviewer 1 Report

Line 302: write "1:20 0,5 McFarland Standard"

For the next paper, it would be better if the Authors evaluated the effect on more strains with different amtimicrobial susceptibility, including also metronidazole, amoxicillin, or levofloxacin resistant strains. 

Reviewer 2 Report

The authors improved the manuscript and therefore the manuscript should be accepted for publication.
